# In Vitro Evaluation of a Nanoparticle-Based mRNA Delivery System for Cells in the Joint

**DOI:** 10.3390/biomedicines9070794

**Published:** 2021-07-08

**Authors:** Lisa Sturm, Bettina Schwemberger, Ursula Menzel, Sonja Häckel, Christoph E. Albers, Christian Plank, Jaap Rip, Mauro Alini, Andreas Traweger, Sibylle Grad, Valentina Basoli

**Affiliations:** 1Institute of Tendon and Bone Regeneration, Spinal Cord Injury & Tissue Regeneration Center Salzburg, Paracelsus Medical University, 5020 Salzburg, Austria; lisa.sturm@pmu.ac.at (L.S.); b.schwemberger@pmu.ac.at (B.S.); 2Austrian Cluster for Tissue Regeneration, 1200 Vienna, Austria; 3AO Research Institute Davos, 7270 Davos Platz, Switzerland; ursula.menzel@aofoundation.org (U.M.); mauro.alini@aofoundation.org (M.A.); valentina.basoli@aofoundation.org (V.B.); 4Department of Orthopaedic Surgery and Traumatology, Inselspital, Bern University Hospital, University of Bern, 3010 Bern, Switzerland; sonja.haeckel@insel.ch (S.H.); christoph.albers@insel.ch (C.E.A.); 5ETHRIS GmbH, 82152 Planegg, Germany; plank@ethris.com; 620Med Therapeutics B.V., Galileiweg 8, 2333BD Leiden, The Netherlands; rip@20medtx.com; 7Department of Health Sciences and Technology, ETH Zurich, 8092 Zurich, Switzerland

**Keywords:** transfection, bioresponsive polymer-based nanoparticles, joint therapies, therapeutic mRNA, biologicals

## Abstract

Biodegradable and bioresponsive polymer-based nanoparticles (NPs) can be used for oligonucleotide delivery, making them a promising candidate for mRNA-based therapeutics. In this study, we evaluated and optimized the efficiency of a cationic, hyperbranched poly(amidoamine)s-based nanoparticle system to deliver tdTomato mRNA to primary human bone marrow stromal cells (hBMSC), human synovial derived stem cells (hSDSC), bovine chondrocytes (bCH), and rat tendon derived stem/progenitor cells (rTDSPC). Transfection efficiencies varied among the cell types tested (bCH 28.4% ± 22.87, rTDSPC 18.13% ± 12.07, hBMSC 18.23% ± 14.80, hSDSC 26.63% ± 8.81) and while an increase of NPs with a constant amount of mRNA generally improved the transfection efficiency, an increase of the mRNA loading ratio (2:50, 4:50, or 6:50 *w*/*w* mRNA:NPs) had no impact. However, metabolic activity of bCHs and rTDSPCs was significantly reduced when using higher amounts of NPs, indicating a dose-dependent cytotoxic response. Finally, we demonstrate the feasibility of transfecting extracellular matrix-rich 3D cell culture constructs using the nanoparticle system, making it a promising transfection strategy for musculoskeletal tissues that exhibit a complex, dense extracellular matrix.

## 1. Introduction

Treatment of musculoskeletal disorders affecting the synovial joint, such as osteoarthritis (OA) [1], gout [2], rheumatoid arthritis [3], scleroderma [4], or systemic lupus erythematosus [5] is extremely challenging and effective therapies remain urgently needed. A particular challenge is posed on the one hand by the complex structure of synovial joints, due to the composition of different tissues, such as bone, cartilage, tendons and ligaments, muscles and synovial membrane, and on the other hand by the tissue-specific properties, such as the extent and types of extracellular matrix [6]. The current pharmacological treatments for OA are often based on steroids (i.e., glucocorticoids) [7] and non-steroidal anti-inflammatory drugs (NSAID) [8], which treat the symptoms but are not curative. The various drugs currently in use are either administered systemically or locally, e.g., by infiltration of glucocorticoids. However, the efficacy of new treatment modalities, such as recombinant proteins, is often hampered by the low penetration of substances into tissues with dense extracellular matrices [9,10]. For example, recombinant proteins such as bone morphogenetic proteins (BMPs) have been employed to enhance bone and cartilage repair. It has been observed, however, that the primary limitation is often the failure of larger molecules to reach deeper regions of the tissue [11]. In addition, cartilage and tendons are not, or barely, vascularized and therefore, typically less amenable to treatments [12,13]. Further, the relatively short half-life and high clearance rates can limit the effectiveness of protein-based therapies with the additional risk of an immune response to exogenously produced proteins [14,15]. In this respect, new effective, targeted, and sustained treatments, which can reach tissues and cells within the joint, or generally in the musculoskeletal system, remain an unmet need.

A more recent class of recombinant proteins, i.e., chimeric proteins, aimed to produce modified and engineered proteins with improved therapeutic function, e.g., a biologically active molecule with a short half-life can be paired with an inactive one, or certain tissues can be targeted by fusing therapeutic or toxic proteins specific for the respective tissue [16,17]. Mouse-human chimeric tumor necrosis factor-alpha (TNFα) blocking molecular antibody (cA2, infliximab) [18] is used to treat inflammatory conditions, like Crohn’s disease [19] and rheumatoid arthritis [20]. Nevertheless, as for conventional drug therapies (e.g., small molecules), exogenous and sometimes repetitive administration is required for all recombinant proteins. To overcome this issue, gene therapies have been developed to enable therapeutic protein production in situ in the affected tissues. The first generation of gene therapies relied on the use of viral vectors carrying DNA encoding the target protein and its subsequent integration into the genomic DNA [21,22]. This type of therapy raised safety concerns due to the impossibility to control intracellular behavior of the vector and its cargo, mostly related to non-specific integration into the genome. The more recent introduction of the CRISPR/Cas technology, where a specific gene can be targeted, modified, turned on/off, has propelled gene therapy into a new era but still results in permanent integration into the genome [23,24]. Therefore, alternative therapies are based on a highly promising technology that uses in vitro-transcribed (IVT) mRNA to deliver genetic information [25,26,27]. Currently, there are approximately 1750 trials for such therapies in various fields, e.g., oncology, cardiology, pneumology, virology [28], and IVT mRNA is currently being employed for the formulation of vaccines against coronavirus disease (COVID-19). However, one drawback of naturally occurring mRNA species is their relative instability and their immunogenicity. Therefore, different strategies have been developed to increase the half-life of chemically produced mRNA therapeutics, e.g., by chemical modification of the 5′-cap and a 3′ poly(A) tail or the use of N1-methyl-pseudouridine (ψ) in the open reading frame (ORF) and untranslated regions (UTRs). In general, the delivery of mRNAs to the cells is considered more effective and safer compared to previously used strategies (e.g., plasmids), since mRNA cannot be genetically integrated and acts in a transient manner [29]. For mRNA-based therapies to be efficacious, an appropriate delivery system is required, which is not only essential for effective delivery of the active oligonucleotide compound but can also impact the pharmacodynamics and pharmacokinetics. A majority of the delivery modalities have made use of various viruses, raising similar concerns as for viral delivery of DNA. Furthermore, viral vectors are known to spread out to other organs, which has already been shown for intra-articular injection [30]. Non-viral delivery of nucleic acids into cells can be achieved with lipids (liposomes) or lipid nanoparticles (LNP) by fusion with the cellular membrane [31,32,33,34]. However, although Lipofectamine (Invitrogen, Waltham, USA) or similar lipid-based reagents achieve high transfection efficiencies in vitro, the in vivo application is mostly not feasible due to their high cell toxicity [35,36]. Another approach makes use of nanoparticles (NPs) as a delivery system. NPs resemble particles, with a diameter of 1–1000 nm in at least one dimension [37], and these non-viral delivery technologies make use of synthetic polymers. Overall, a suitable NP-based transfection method should demonstrate minimal cytotoxicity, result in high expression levels of the gene-of-interest, and be highly reproducible.

In this study, we used polymeric hybrid constructs based on multifunctional poly(amidoamines), previously developed as nanoparticle carriers for intracellular delivery of (oligo-) nucleotides [38,39,40]. The previously described linear poly(amidoamines) were used as a polyplex for delivery of oligonucleotides. For the current study, the polymers were hyperbranched and cross-linked to further increase the delivery efficacy and the stability of the nanoparticles, with the aim to create a polymeric network that can be cross-linked to form a particle. The cationic core of the nanoparticles allows efficient loading of mRNAs by electrostatic interactions in aqueous solution without the need for proprietary instrumentation. The use of these nanoparticles further allows scale-up as the materials can be produced in bulk and formulated with any desired oligonucleotide payload for different applications.

The objective of this study was to assess and optimize the nanoparticle: mRNA formulation for effective mRNA delivery to primary cells present in joints. Local delivery of loaded nanoparticles into the articular joint allows for targeting negatively charged cells and tissues such as cartilage or tendon, potentially enhancing mRNA delivery and treatment efficacy for joint-related diseases. Therefore, we used various cell types from different species: human bone marrow stromal cells (hBMSC) and synovial derived stem cells (hSDSC), which are cells commonly used in orthopedic tissue engineering approaches; bovine chondrocytes (bCH), an abundant source of cells commonly used for studying cartilage biology in 2D and 3D in vitro models; and rat tendon derived stem/progenitor cells (rTDSPC), as rats are frequently used for studying tendon regeneration, allowing translation of in vitro results to in vivo studies. Together, we demonstrate that the presented NP-based delivery system has potential for mRNA-based treatment of musculoskeletal and joint-related pathologies, paving the way for future pre-clinical investigations.

## 2. Materials and Methods

### 2.1. Nanoparticle Synthesis

Cystamine bis(acrylamide) was synthesized from acryloyl chloride (Sigma-Aldrich, St. Louis, MO, USA) and cystamine HCl (Sigma-Aldrich, St. Louis, USA) as described previously by Lin et al. [41]. Poly(amidoamine)s were synthesized by Michael-type polymerization of primary (or bis-secondary) amines linked with bis-acrylamides as described previously by Lin et al. [41]. Branched PEI800 (2% *w*/*w*, Sigma-Aldrich, St. Louis, MO, USA) was added to the reaction mixture and hyperbranched poly(amidoamine) was made. This hyperbranched polymer was dissolved in water with photoinitiator Irgacure 2959 (Sigma-Aldrich, St. Louis, MO, USA) and emulsified in mineral oil (VWR chemicals, Radnor, PA, USA) with 10% of surfactant ABIL EM 90 (Sigma-Aldrich, St. Louis, USA) using ultrasonication. The emulsion was exposed to UV irradiation to crosslink polymers and form the NPs. The NPs were dialyzed against water, freeze-dried, and stored in the freezer. The size and zeta potential of the NPs were measured using Dynamic Light Scattering (DLS) (Zetasizer Nano ZS, Malvern, Kassel, Germany).

The encapsulation of mRNA into NPs was verified with an agarose gel electrophoresis. Therefore, free mRNA, 10 µL of 1.5 mg/mL NPs complexed with mRNA (2:50 *w*/*w* loading ratio), and loaded NPs treated with 2 M 1,4-Dithiothreitol (DTT) containing 5 mg/mL heparin were heated for 5 min at 60 °C and loaded on a 1% agarose gel. Electrophoresis ran at 90 mA for 15 min.

### 2.2. mRNA Synthesis

Chemically modified mRNA (mRNA) was synthesized as previously described by Kormann et al. [42]. In brief, the respective pDNA templates were subjected to in vitro transcription using T7 RNA Polymerase (Thermo Fisher Scientific, Waltham, MA, USA) with a predefined mix of natural and chemically modified ribonucleotides. To enhance RNA stability, a m7G cap structure was added to the 5′ end of the transcript, while the 3′ end was subjected to enzymatic polyadenylation of ~120 nucleotides. The mRNA product was purified by ammonium-acetate precipitation and was re-suspended in water at the desired concentration. Standard 260/280 nm ratio was determined on a NanoDrop2000C spectrophotometer (Thermo Fisher Scientific, Waltham, USA) and mRNA integrity and size were determined using a Standard Sensitivity RNA Analysis Kit on a Fragment Analyzer (Advanced Analytical Technologies, Ankeny, IA, USA).

### 2.3. Cell Isolation

#### 2.3.1. Bovine Chondrocytes (bCH) Isolation

Metacarpal joints of 6–12 months old calves were acquired from a local abattoir. Joints were opened, cartilage chips were removed from subchondral bone and transferred into prepared tubes containing sterile phosphate-buffered saline (PBS) with 10× penicillin/streptomycin (P/S) (Gibco by Life Technologies, Waltham, MA, USA, #15140-122). Cartilage chips were transferred into spinner flasks and washed three times for 15 min each with PBS containing 1× P/S. Tissues were predigested for 1 h using 0.1% Pronase (Roche, Basel, Switzerland; in DMEM + 1× P/S, sterile filtered), and then incubated with 600 U/mL collagenase II (Worthington Biochemical) for 14 h at 37 °C, 5% CO_2_, 90% humidity. The next day, Dulbecco’s Modified Eagle Medium (DMEM, Gibco by Life Technologies, Waltham, MA, USA, #52100-021 + 3.7 g/L NaHCO_3_ + 0.1 g/L sodium pyruvate) containing fetal bovine serum (FBS, Gibco by Life Technologies, Waltham, MA, USA, #10500) was added to a final concentration of 10% FBS in the digested tissue solution. The contents of the spinner flask were passed through a cell strainer (40 µm) into centrifugation tubes and centrifuged at 565× *g* for 7 min. The pellet was resuspended and washed two more times with complete medium. Cells were counted in a Neubauer cell chamber and seeded at 3000 cells/cm^2^. Bovine chondrocytes were grown in DMEM (Gibco by Life Technologies, Waltham, MA, USA, #52100-021 + 3.7 g/L NaHCO_3_ + 0.1 g/L sodium pyruvate), 10% FBS, 1× P/S.

#### 2.3.2. Isolation of Human Mesenchymal Stem Cells (hBMSC)

Bone marrow aspirates were obtained from patients undergoing spine surgery at the Inselspital Bern. The Swiss Human Research Act does not apply to research that involves anonymized biological material and/or anonymously collected or anonymized health-related data. Therefore, this project did not need to be approved by the ethics committee. General Consent which also covers anonymization of health-related data and biological material was obtained. BMSC isolation was performed via Ficoll density gradient centrifugation as described in Rothweiler et al. [43]. Human BMSC were cultured in BMSC medium consisting of Minimum Essential Medium α-modification (Gibco by Life Technologies, Waltham, MA, USA, #1200-063 + 2.2 g/L NaHCO_3_), 10% FBS (Sera Plus, PAN-Biotech, Aidenbach, Germany, #3702-P121812), 1× P/S (Gibco by Life Technologies, Waltham, MA, USA, #15140-122), 5 ng/mL basic fibroblast growth factor (bFGF; Fitzgerald, Acton, MA, USA, #30R-AF015).

#### 2.3.3. Isolation of Human Synovial-Derived Stem Cells (hSDSC)

Synovial membranes were obtained from patients after informed consent was given (ethical approvals GTE-2002 and AN-EK-FRBRG-135/14). Cell isolation was performed according to Kovermann et al. [44]. SDSC growth medium consisted of DMEM, 10% FBS (Sera Plus, PAN-Biotec, Aidenbach, Germany), 1× P/S, 5 ng/mL bFGF.

#### 2.3.4. Isolation of Rat Tendon Derived Stem and Progenitor Cells (rTDSPC)

Male Sprague Dawley rats (12 weeks old) were euthanized with an intracardiac injection of pentobarbital under deep isoflurane anesthesia. Rat tendon-derived stem and progenitor cells (rTDSPC) were isolated as follows: Achilles tendons were dissected under sterile conditions, cut into small pieces, and digested overnight at 37 °C in 3 mg/mL collagenase type II (Gibco by Life Technologies, Waltham, MA, USA, #17101015) in αMEM medium (Sigma-Aldrich, St. Louis, MO, USA, #M4526) containing 10% FBS (Sigma-Aldrich, St. Louis, MO, USA, #F7524), 1× GlutaMax (Gibco by Life Technologies, Waltham, MA, USA, #35050-038) and 1× P/S (Sigma-Aldrich, St. Louis, MO, USA, #P4333), subsequently referred to as αMEM complete medium. The isolated cells were collected on the following day, washed with complete medium and plated on a 175 cm^2^ cell culture flask. At 70–80% confluency, cells were collected and prepared for cryopreservation according to standard protocols.

### 2.4. Cell Culture

All cells were grown under standard cell culture conditions (37 °C, 5% CO_2_, 90% humidity). Medium changes were performed every 2–3 days and cells were passaged at 80% confluency using Trypsin-EDTA (Sigma-Aldrich, St. Louis, MO, USA). For the transfection experiments, cells were used up to passage 4 and seeded at 15,000 cells/cm^2^ using their described respective growth medium. All experiments were performed in monolayer culture in TC24 well plates with an area of 1.92 cm^2^ per well.

### 2.5. 3D Cultures

BCH pellets were created and cultured in chondrogenic medium consisting of DMEM-HG, 1× ITS+ (Corning, Corning, NY, USA, #354352), 1× non-essential amino acids (Gibco by Life Technologies, Waltham, MA, USA, #11140-035), 1× P/S, 50 µg/mL ascorbic acid 2-phosohate (Sigma, #A8960), 10^−7^ M dexamethasone with 10 ng/mL transforming growth factor β (TGFβ; Fitzgerald, Acton, MA, USA, #30R-AT072). Briefly, monolayer expanded bCH were detached and transferred into a 96-well V-bottom plate (Thermo Fisher Scientific, Waltham, MA, USA, #612V96 + #64200) at a density of 0.2 × 10^6^ cells/well. Plates were centrifuged at 400× *g* for 5 min to collect cells at the bottom of the plates. Medium changes were performed every 2–3 days. Pellets were cultured for 7 days before transfection.

Tendon-like constructs of rTDSPC were created according to Gehwolf et al. [45] and transfected after 7 days of culture. Z-stacks of unfixed pellets and unfixed constructs were acquired 24 h after transfection (LSM 800/710, Zeiss, Oberkochen, Germany). All transfection experiments were performed in triplicate.

### 2.6. Delivery of mRNA Using Nanoparticles

Polymeric nanoparticles (NPs; 20Med Therapeutics, Leiden, The Netherlands) were supplied in lyophilized form and were freshly reconstituted in 20 mM sterile HEPES buffer (Gibco by Life Technologies, Waltham, MA, USA, #15630-056) at 3 mg/mL just prior to the experiment. A solution of mRNA (A) was prepared in 20 mM sterile HEPES buffer with the appropriate concentration and an equal volume of nanoparticle solution (B) was set up. Then, the mRNA solution (A) was mixed with an equal volume of NP solution (B) and the resulting solution (A + B) was incubated at room temperature for 15 min. Complete growth medium appropriate for the respective cell type was added and NPs were delivered to the cells in 24-well plates in a total volume of 0.6 mL/well. Imaging, viability testing, and flow cytometry were performed 24 h after transfection of the cells with mRNA-loaded NPs. Experiments were performed using chemically modified, ARCA capped mRNA encoding tdTomato at the indicated concentrations. mRNAs were provided by Ethris GmbH (Planegg, Germany).

### 2.7. Fluorescence Imaging and Live/Dead Staining

Transfection efficiency of tdTomato mRNA loaded NPs was evaluated by laser confocal microscopy (Zeiss, Oberkochen, Germany, LSM700/710 or LSM 800) after 24 h. Transfected cells were co-stained with 5 µM Calcein (Sigma-Aldrich, St. Louis, MO, USA, #17783) and 0.625 µg/mL DAPI (Sigma-Aldrich, St. Louis, MO, USA, #D9542) for 30 min to discriminate between live and dead cells [46].

### 2.8. Metabolic Activity and Toxicity Screening

AlamarBlue assay (BioRad, Hercules, CA, USA, #BUF012A) was used to determine metabolic activity 24 h after transfection. The medium was aspirated and cells were washed once with PBS. AlamarBlue solution was prepared according to the manufacturer’s instructions using the respective growth medium for each cell type and 100 µL were added per well. After 4 h under standard cell culture conditions, the medium was transferred into a 96-well white plate and the fluorescent signal was measured at λ_ex_560/λ_em_590 nm. Working solution only was used as minimally viable control. Wells were blanked with average medium fluorescence and metabolic activity was determined by the percentage of control cells treated only with HEPES buffer.

### 2.9. Flow Cytometry

Cells were washed with PBS, 100 µL/cm^2^ Trypsin-EDTA was added to each well and incubated for 4 min at 37 °C. Cells from three technical replicates were pooled into round-bottom tubes and centrifuged for 5 min at 400× *g*. The medium was aspirated, the pellet was resuspended in 100 µL flow cytometry buffer (3% serum in PBS) and passed through a 35 µm cell strainer. Tubes were spun briefly to pass the cells through the strainer and kept on ice until measurement was performed (BD FACSAria III Cell Sorter, BD Accuri C6 Plus Flow Cytometer, Franklin Lakes, NJ, USA). Forward, side scatter, and laser voltage were adjusted using untreated cells. Cells treated with unloaded NPs were used as a reference.

### 2.10. Statistics

Statistical analyses were performed using Graph Pad Prism software (version 5.04; GraphPad Software, San Diego, CA, USA). One-way analysis of variation (ANOVA) applying the Dunnett’s Multiple Comparison test were used to test for differences between the metabolic activity of the treatment groups, using untreated cells as control. Differences in transfection efficiencies were analyzed using One-way analysis of variation (ANOVA) applying the Tukey’s Multiple Comparison test or the Dunnett’s Multiple Comparison test when testing against the lowest NP:mRNA-treatment group as a reference. A *p*-value < 0.05 was considered statistically significant.

## 3. Results

### 3.1. Nanoparticle Characterization

Ten µL of 1.5 µg/mL NPs loaded with mRNA (2:50 *w*/*w* ratio) were analyzed on an agarose gel to confirm the encapsulation of mRNA. Loading mRNA in the NPs prevents migration in the agarose gel. Samples were treated with 2 M DTT containing 5 mg/mL heparin and incubated for 5 min at 60 °C to release the mRNA from the particles, which was shown by migration of released mRNA in the agarose gel (Figure 1). The average NP size determined by DLS was 82 ± 4 nm (Figure 2), the polydispersity index was 0.170 ± 0.009, and the zeta potential 32 ± 3 mV (*n* = 3).

### 3.2. Nanoparticle-Based mRNA Delivery to Different Cell Populations of Articular Joints

An initial experiment was performed, where a standard amount of 480 ng mRNA encoding for tdTomato was complexed with 12 µg NPs (2:50 *w*/*w* ratio mRNA:nanoparticle) and delivered in a 24-well format to different musculoskeletal cell types; bCH, rTDSPC, hBMSC, hSDSC, respectively (Figure 3). After 24 h, tdTomato mRNA-loaded NPs were taken up by all tested cell types and resulted in a translation of the mRNA into functional, fluorescent tdTomato, while NPs or mRNA alone did not result in any fluorescent signal. Live/dead cell staining with calcein (live) and DAPI (dead) showed that the nanoparticle-based delivery of mRNA did not impair the viability of cells (Figure 3A). Flow cytometry analysis demonstrated a transfection efficacy of approximately 20% for bCH (28.40% ± 22.86), rTDSPC (18.13% ± 12.07), hBMSC (18.23% ± 14.80), and hSDSC (23.63% ± 8.81) (Figure 3B). The metabolic activity of the cells was determined using an AlamarBlue assay 24 h after treatment of the cells. Neither the complexed NPs nor the NPs alone had a negative effect on the metabolic activity of bCH (103.55% ± 10.93), rTDSPC (103.28% ± 0.546), and hBMSC (84.82% ± 13.54), while hSDSC (57.65% ± 24.48) showed a significant decrease in their metabolic activity (*p* < 0.05, Dunnett’s Multiple Comparison test) when treated with 12 µg NPs only (Figure 3C).

### 3.3. Effect of Different mRNA:Nanoparticle Ratios on Transfection Efficiency and Cell Viability in bCH and rTDSPC

To examine if different concentrations of NPs (with a constant mRNA amount) could improve the transfection efficiency of bCH and rTDSPC without affecting their viability, 480 ng tdTomato mRNA was complexed with 6, 12, 24, and 36 µg NPs in 0.6 mL, resulting in mRNA:nanoparticle *w*/*w* ratios of 2:25, 2:50, 2:100, and 2:150, respectively. As shown in Figure 4A, increasing the amount of NPs resulted in an increased expression of tdTomato, although the same amount of mRNA was used for complex formation. bCH showed a significant increase of transfection efficiency when 12 µg of complexed NPs were used (41.07% ± 19.68) in comparison to 6 µg NPs (9.47% ± 5.00; *p* < 0.05, Dunnett’s Multiple Comparison test). For rTDSPC, transfection efficiency was highest when 24 µg of complexed NPs were used (39.27% ± 8.22; Figure 4B). A higher NP amount seemed detrimental, as transfection efficiencies generally decreased. Further, a significant reduction of metabolic activity was also evident when 24 µg (bCH 49.58% ± 13.22, rTDSPC 78.91% ± 17.79) or more of the complexed NPs were used for mRNA delivery (*p* < 0.01 for bCH, *p* < 0.05 for rTDSPC; Dunnett’s Multiple Comparison test), indicating a cytotoxic effect (Figure 4C). In particular, 36 µg of complexed NPs resulted in significant cell death (bCH 26.5% ± 15.08; rTDSPC 22.91% ± 0.30; *p* < 0.001, Dunnett’s Multiple Comparison test), as also indicated by the bright-field microscopy.

### 3.4. Transfection Efficiency and Metabolic Cell Activity after Treatment with Increasing Amounts of mRNA Complexed to 6 or 12 µg of Nanoparticles

As 6 and 12 µg of NPs resulted in no noticeable cytotoxicity, we next determined the optimal mRNA:nanoparticle loading ratio in bCH and rTDSPC. Then, 6 or 12 µg of NPs were complexed with an increasing amount of tdTomato-mRNA, resulting in a 2:50, 4:50, and 6:50 *w*/*w* loading ratio of mRNA:NPs. As expected, 6 µg of complexed NPs generally resulted in a low number of tdTomato-positive cells, with only a minimal effect observed for the increasing mRNA loading ratios (Figure 5A). This was also confirmed by flow cytometry analysis, demonstrating a higher transfection efficacy when 12 µg of NPs were used. Interestingly, the different mRNA ratios did not improve the transfection efficiency. Overall, bCH showed a better transfection efficacy when compared to rTDSPC (Figure 5B). The metabolic activity of bCH and rTDSPC was not affected by any of the different mRNA:NP loading ratios tested (Figure 5C).

### 3.5. Treatment of bCH and rTDSPC with the Optimal mRNA:Nanoparticle Loading Ratio of 2:50

The previously determined optimal complexing ratio of 2:50 (mRNA:NPs) was used to further optimize the transfection efficiency for both cell types. Therefore, cells were incubated for 24 h with an increasing absolute amount of mRNA-complexed nanoparticles, ranging from 6, 12, 24 to 36 µg. By increasing the amount of loaded NPs, the transfection efficiency could significantly be improved, particularly for rTDSPC (Figure 6A,B). Interestingly, differences between the two cell types were observed for their metabolic activity. Whereas no signs of cytotoxicity and good transfection efficiency were seen for rTDSPC if treated with 24 µg of nanoparticles complexed with 960 ng of mRNA encoding tdTomato (viability 88.59% ± 19.89), a significant drop in activity of bCH treated with this amount of NPs (54.17% ± 32.56; *p* < 0.05, Dunnett’s Multiple Comparison test) was evident (Figure 6C). For both cell types, treatment with 36 µg of complexed NPs was detrimental (bCH 13.99% ± 11.66, rTDSPC 39.41% ± 12.39; *p* < 0.001, Dunnett’s Multiple Comparison test). As also a loss in metabolic activity was observed when 36 µg of unloaded NPs were used (bCH 36.17% ± 13.55, rTDSPC 45.56% ± 16.52; *p* < 0.001, Dunnett’s Multiple Comparison test), a cytotoxic effect elicited by the mRNA or the tdTomato protein can be excluded. Proof-of-concept experiments using the mRNA:NPs ratio of 2:50 in different absolute amounts for treatment of 3D constructs showed successful transfection of cells embedded in a more complex matrix (Figure 7). An increased tdTomato signal was observed with increased particle dose, which is in line with results obtained in 2D experiments.

## 4. Discussion

Our study showed that the used NPs can be employed for the intracellular delivery of (oligo-) nucleotides. Moreover, we optimized the transfection efficiency by keeping the NPs at a low concentration, while improving the amount of loaded mRNA.

Recently, new and promising modalities, such as therapeutic mRNAs, have been proposed that could effectively allow the production of proteins in situ, eliciting an effective physiological response [42]. To improve mRNA-delivery into target cells and tissues, we used poly(amidoamine)-based NPs for the intracellular delivery of (oligo-) nucleotides (Figure 1). NPs were improved for in vivo application by branching and cross-linking the poly(amidoamines) [38]. The mRNA is taken up by the cells based on proton buffering functionalization during loaded NPs formation, and adhesion of resulting stabilized nanoparticle-mRNA on the cell membrane that permits the release into the cytosol. Subsequently, as previously described by Philipps et al., due to the relatively high concentration of glutathione in the cytosol, the disulfide bonds in the nanoparticle network become rapidly cleaved by the thiol-disulfide exchange reaction, resulting in rapid disassembly of the nanoparticle-cage, release of the mRNA payload into the cytosol, and subsequent protein translation [47].

Generally, NPs can be broadly applied for oligonucleotide delivery, but it is of paramount importance to optimize them specifically for the target cells and tissues with different phenotypes. By using a set of primary cell types present in a synovial joint, we observed that every cell type had a distinct capability to take up the NPs and/or translate mRNA to protein; in our case, we used mRNA encoding for tdTomato, a trackable red fluorescent translated protein, to evaluate the yield and efficiency of transfection. The results demonstrate that although we used the same amount of mRNA in chondrocytes, tendon-derived stem/progenitor cells, synovium stem cells, and bone marrow stem cells, albeit from different species, the transfection and yield of tdTomato protein expression were different among all cell types investigated (Figure 3). This could be explained by the varying ability and extent to which extracellular matrix is produced in 2D culture, which could interfere with the delivery of mRNAs using NPs [48,49,50]. Furthermore, we sought to overcome this limitation by optimizing parameters like the quantity of mRNA, NPs, and the effective ratio between them. With this aim, we evaluated the effect of increased NP concentration during transfection while keeping the mRNA amount constant. We focused our attention mainly on chondrocytes and tenocytes which are known to produce high levels of collagen-rich extracellular matrix. Increased concentration of NPs (24 µg; *w/w* ratio 2:100) showed a beneficial effect on the transfection yield; however, a concomitant increase in cytotoxicity was also evident (Figure 4). We hypothesize that the NPs may become toxic on monolayer cells at high concentration due to the high binding or possible interference with cationic charge on the cellular membrane. Subsequent membrane lysis and mitochondrial or lysosomal damage have also been described in literature [51].

Although a high concentration of NPs theoretically allows greater mRNA uptake and protein expression, the NPs themselves may significantly disturb the cell behavior. Thus, we optimized the transfection efficiency by keeping the NPs at a constant low concentration, while increasing the amount of loaded mRNA. As we did not observe an increase in reporter protein production (Figure 5), we hypothesize that the NPs have a saturation level beyond which it is no longer possible to load more nucleic acids. We therefore evaluated how increasing the total amount of complexed NPs (ratio 2:50) impacts transfection efficiency and hence the expression of tdTomato (Figure 6). Thereby, we demonstrated that it was possible to introduce a higher amount of nucleic acids into those cells that produce a high amount of positively charged matrix, resulting in an increased amount of tdTomato expression, by using one single-dose treatment. Successful transfection of bCH in pellets and rTDSPC in tendon-like constructs indicates that more abundant extracellular material did not interfere with the delivery system (Figure 7) and a dose-dependent effect like it was seen in monolayer cultures was also apparent. Hence, we conclude delivery of NPs is feasible in native tissues rich in extracellular matrix.

### 4.1. Strength and Weakness of the Study

The developed NPs and mRNAs proved to be functional at optimized concentrations in a 2D-cell culture setting. The transfection experiments performed on three-dimensional constructs can be regarded as initial indicators that a richer matrix does not inhibit transfection. Further analysis using different methods will be necessary to evaluate performance of the NP delivery system in a 3D environment. Demonstration of the ability of the mRNA:NP complexes to penetrate the ECM and transfect cells embedded in their native matrices is necessary to evaluate if the current formulation will also achieve satisfactory transfection in organotypic culture or in vivo. Moreover, further evaluation of the NPs in terms of biocompatibility and performance in vivo will be needed. However, the general feasibility and optimization of loading ratios and doses shown by the in vitro data presented in this study serve as an important basis for future experiments in 3D-in vitro/ex vivo and in vivo settings. The high amount of matrix present in the tissues of interest, i.e., cartilage and tendon, may help to further attenuate toxicity and will possibly allow the use of higher NP dosages to increase transfection performance. This has been shown for other nanomaterials [52], cytotoxic agents [53], radiation [54] in 3D culture experiments, and for other polymeric NPs in vivo [55].

### 4.2. Clinical Outlook

Drug delivery by NPs has several advantages: good penetration of biological membranes, the possibility of targeted drug delivery, improved solubility of hydrophobic drugs, and increased drug stability. In addition, the use of nanoparticle, compared to other methods like Lipofectamine, offers the possibility to be freeze-dried and remain stable over prolonged time periods. Furthermore, due to the nature of the material, specific optimization for the type of targeted cells or tissue is possible to provide high performance during transfection. In vitro studies have shown that NPs below 90 nm penetrate cartilage explants through their entire thickness, while larger NPs tend to accumulate in the superficial layers [37]. In vivo experiments in healthy rats also showed that avidin, a NP carrier, penetrates throughout articular cartilage within six hours and can remain there for up to one week [56]. Indeed, it has been shown that the concentration of endogenous hyaluronate in the synovial fluid is sufficient to associate with NPs and slow their otherwise rapid clearance from the synovium [57]. Despite all these advantages, there are also some disadvantages. For example, NPs can aggregate, they can be immunogenic, and the solvent of the particles can cause cell toxicity [37]. Overall, however, it can be stated that the clinical application of NPs is still in its infancy, especially in relation to the application of musculoskeletal diseases. Although there are some FDA-approved NP therapies and several ongoing clinical trials, these are mainly investigating the treatment of diseases such as cancer, infectious diseases, and vaccination or iron deficiency using NPs. Currently, there are no clinical trials investigating the administration of NP for the treatment of musculoskeletal diseases, particularly joint diseases such as OA [58].

Regarding the treatment approach with mRNA, in vivo studies have already shown that intra-articular injection of nanomicelles carrying runt-related transcription factor 1 (RUNX1) mRNA increased cartilage anabolic markers and exogenous protein expression in OA mouse models. The authors concluded that the progression of OA was greatly slowed by the in situ delivery of RUNX1 mRNA [59], indicating that IVT mRNA therapy is a promising approach to treat OA and other pathologies of the musculoskeletal system.

## 5. Conclusions

By keeping a low and well-defined ratio between mRNA and NPs, it is possible to introduce nucleic acids in cells that produce a positively charged matrix, preserving their quality, while inducing the translation of mRNA in complex proteins. Moreover, NP transfection proved successful for various cell types present in the musculoskeletal system and may be a promising method for tissues that exhibit a complex dense matrix. Our observation and optimization of the NPs and mRNA dose and ratio will pave the way for further applications. Local NP-based delivery of therapeutic mRNAs for the modulation of OA will be further studied in vitro and in vivo.

## Figures and Tables

**Figure 1 biomedicines-09-00794-f001:**
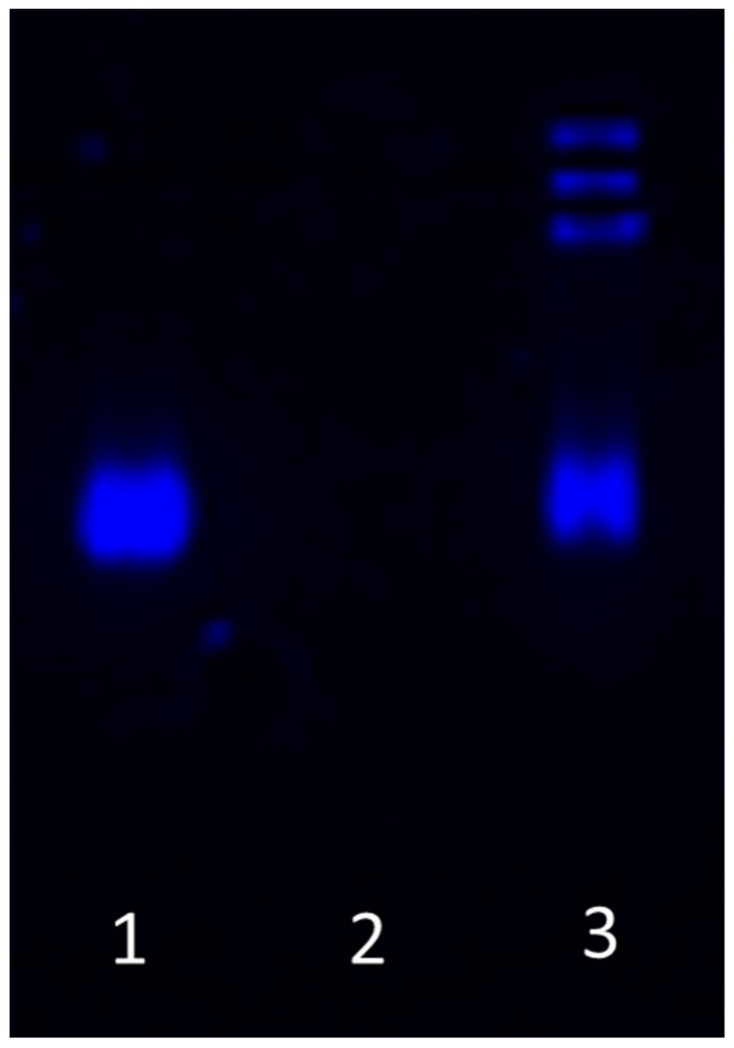
Migration of mRNA in agarose gel. (**1**) Free mRNA; (**2**) mRNA loaded nanoparticles; (**3**) mRNA released from nanoparticles with 5 M DTT containing 5 mg/mL heparin.

**Figure 2 biomedicines-09-00794-f002:**
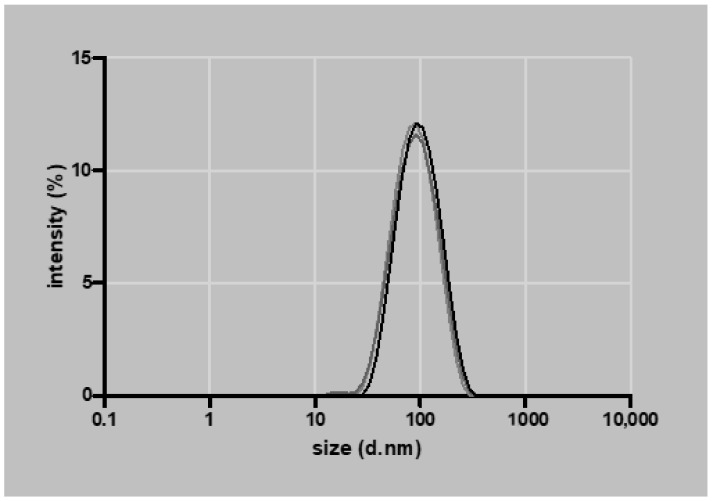
Particle size distribution measured by % intensity and particle diameter expressed in nanometer (d.nm; *n* = 3).

**Figure 3 biomedicines-09-00794-f003:**
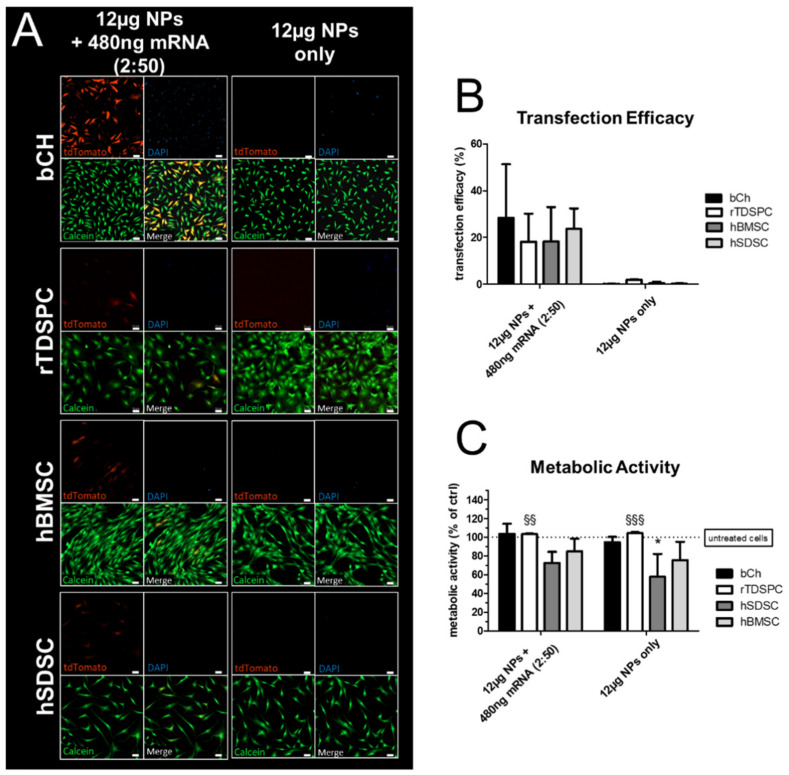
Cell viability and transfection efficiency of different primary cells treated with tdTomato mRNA-complexed nanoparticles (NPs). (**A**) Fluorescence images of bCH, rTDSPC, hBMSC, hSDSC treated with 12 µg NPs complexed with 480 ng tdTomato mRNA (2:50 *w*/*w* ratio mRNA:NPs). tdTomato-positive cells (red), dead cells stained with DAPI (blue), and live cells stained with Calcein (green). Scale bar = 50 µm. (**B**) Transfection efficiency analyzed with flow cytometry shown as % tdTomato-positive cells of the entire cell population (*n* = 3; ANOVA and Tukey’s Multiple Comparison test). (**C**) Metabolic activity is shown as % metabolic activity relative to control cells treated with HEPES buffer (*n* = 3; ANOVA and Dunnet’s Multiple Comparison test using untreated cells as control). ^§^ used to indicate significance for rTDSPC, * for hSDSC; * *p* < 0.05, ^§§^
*p* < 0.01, ^§§§^
*p* < 0.001.

**Figure 4 biomedicines-09-00794-f004:**
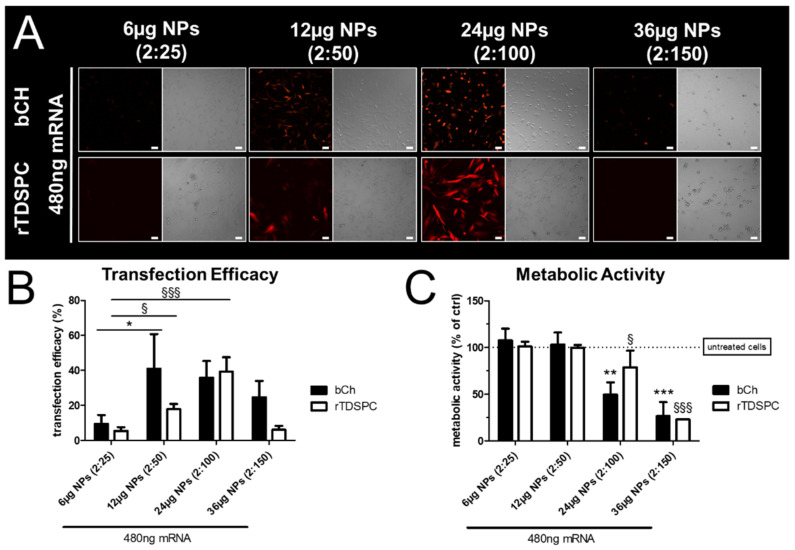
Transfection efficiency and metabolic activity after treatment with varying amounts of nanoparticles and constant mRNA. (**A**) Fluorescent microscopy and bright-field images of cells treated with 6, 12, 24, or 36 µg nanoparticles (NPs) complexed with constant 480 ng tdTomato mRNA. Scale bar = 50 µm. (**B**) Transfection efficiency analyzed with flow cytometry shown as % tdTomato-positive cells of the total population (ANOVA with Dunnet’s Multiple Comparison test using 6 µg NPs [2:25] as a reference; *n* = 3). (**C**) Metabolic activity determined by AlamarBlue assay. Shown is % metabolic activity relative to control cells treated with HEPES buffer (ANOVA with Dunnet’s Multiple Comparison test using untreated cells as reference; *n* = 3). ^§^ used to indicate significance for rTDSPC, * for bCH. */^§^
*p* < 0.05, ** *p* < 0.01, ***/^§§§^
*p* < 0.001.

**Figure 5 biomedicines-09-00794-f005:**
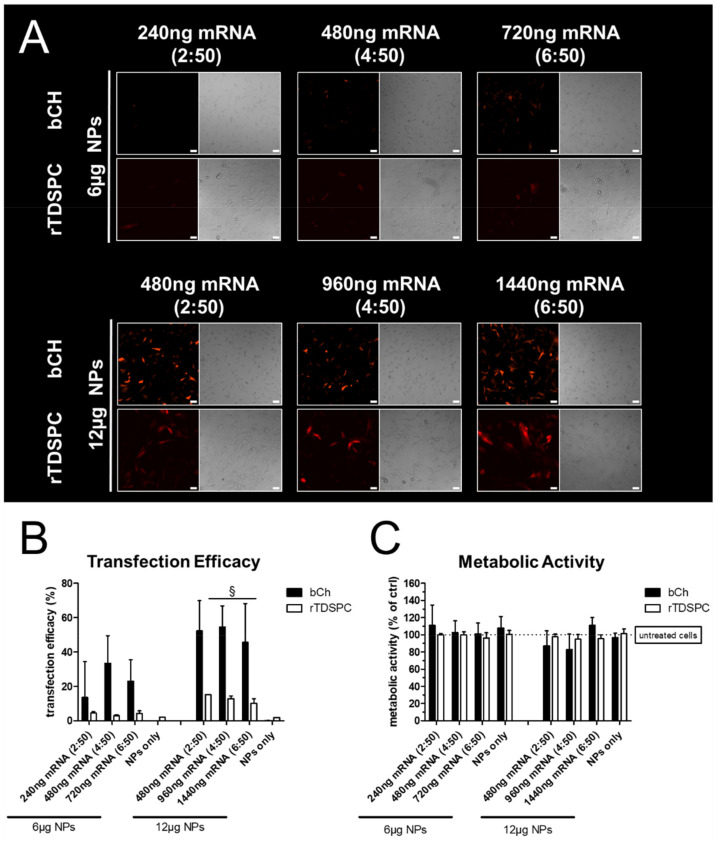
Optimization of mRNA:nanoparticle (NP) loading ratio (*w*/*w*). (**A**) Fluorescence and bright-field images of bCH and rTDSPC treated with 6 or 12 µg nanoparticles and an mRNA loading ratio of 2:50, 4:50, or 6:50. Scale bar = 50 µm. (**B**) Transfection efficiency analyzed with flow cytometry shown as % tdTomato-positive cells of the total population (ANOVA with Tukey’s Multiple Comparison test; *n* = 3). (**C**) Metabolic activity shown as % relative to HEPES-treated control cells. (ANOVA with Dunnet’s Multiple Comparison test using untreated cells as control; *n* = 3). ^§^
*p* < 0.05.

**Figure 6 biomedicines-09-00794-f006:**
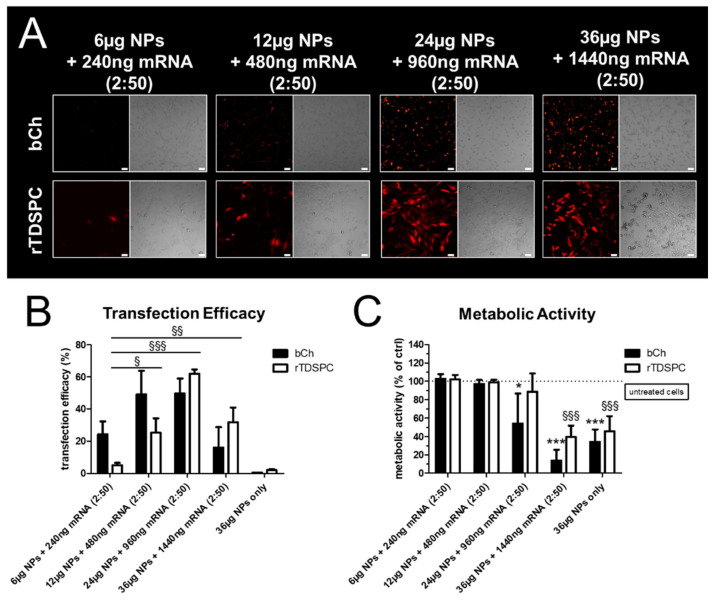
Treatment of bCH and rTDSPC with increasing amounts of the optimal mRNA:nanoparticle loading ratio 2:50. (**A**) Fluorescence and bright-field images of bCH and rTDSPC treated with the different formulations. Scale bar = 50 µm. (**B**) Transfection efficiency analyzed with flow cytometry shown as % tdTomato-positive cells of the total population (ANOVA with Dunnet’s Multiple Comparison test using 6 µg NPs [2:25] as a reference; *n* = 3). (**C**) Metabolic activity determined by Alamar Blue assay (ANOVA with Dunnet’s Multiple Comparison test using untreated cells as reference; *n* = 3). ^§^ used to indicate significance for rTDSPC, * for bCH. */^§^
*p* < 0.05, ^§§^
*p* < 0.01, ***/^§§§^
*p* < 0.001. (*n* = 3).

**Figure 7 biomedicines-09-00794-f007:**
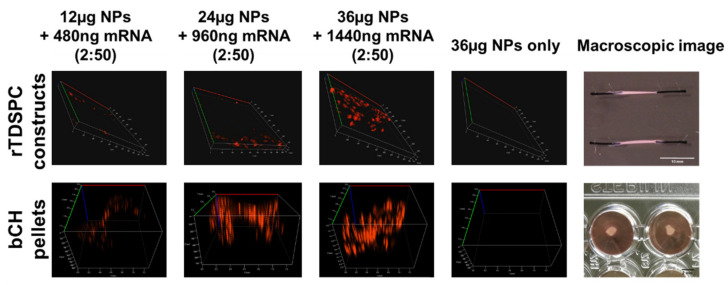
Transfection of 3D constructs of bCH and rTDSPC with the optimal mRNA:nanoparticle ratio of 2:50 (*w*/*w*). Representative Z-stacks of unfixed pellets and 3D tendon-like constructs demonstrating ability of the mRNA-loaded NPs to transfect cells embedded in a dense, collagenous matrix.

## Data Availability

All data generated or analyzed during this study are included in this published article.

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
