# Peer review of "In Vitro Evaluation of a Nanoparticle-Based mRNA Delivery System for Cells in the Joint"

_biomedicines, 2021, doi:10.3390/biomedicines9070794_

Round 1
Reviewer 1 Report
This paper examines the use of novel polymeric nanoparticles for delivering mRNA to a variety of cells found in bone and cartilage. The paper is interesting, but it needs to be placed in the proper context. Please see below for suggestions for improvement.
It would be useful to define what is excellent in terms of transfection efficiency when it is mentioned in the Introduction.
The Introduction also refers to the use of nanoparticles as a newly developed technology even though this technology has been around for awhile. I would suggest rephrasing.
What is the benefit to using hyperbranched polymers?
What is meant by low cost? This would be helpful to quantify.
It would be useful to include the rationale for the choice of cells used and the species used in this study.
In Section 2.2, how was purity quantified? I would add specifics.
In Section 2.5, what does 1+1 mean?
In Figure 1, was a DNA ladder run on the gel for reference? I would include it if possible.
What is the n for Figure 2?
Were any statistical differences observed in Figure 3B and C? Same for Figure 4B and C? And Figure 5B and C? Figure 6B and C?
It would be helpful to include how this method of transfection compares to current methods of transfection. What are the specific advantages of this system?
It would be helpful to discuss the biocompatibility of these materials for clinical translation.
Reviewer 2 Report
Title: In vitro evaluation of a nanoparticle-based mRNA delivery system for cells in the joint
ID: biomedicines-1252452
Authors: Lisa Sturm, Bettina Schwemberger, Ursula Menzel, Sonja Häckel, Christoph E Albers, Christian Plank, Jaap Rip, Mauro Alini, Andreas Traweger, Sibylle Grad and Valentina Basoli
General Comments: In this study, the authors present in vitro data showing the transfection efficiency of mRNA-loaded nanoparticles with regards to the cells that make up the joint including cartilage, tendon and synovial cells with the goal of assessing these NPs as a potential therapeutic carrier for intra-cellular delivery of oligonucleotides. They also attempt to optimize the delivery module by exploring different ratios of mRNA and NP. The authors show that while increasing NP increased the transfection efficiency, this came at the cost of cell metabolic activity and viability for higher amounts. On the other hand, increasing the mRNA amount while keeping NP amount low did not improve transfection efficiency. The optimal combination of 2:50 mRNA to NP was found to be most effective when used in higher total amount with no observed cytotoxicity. While the imaging data provides ample support to the authors’ claims, it must be noted that the analytical data do not have statistics (n for bCH cells is 2 in all figures). Providing statistically analyzed data would greatly improve the strength of the manuscript. This reviewer appreciates the authors clearly listing the shortcomings of their study. It would be interesting to see how this delivery system works in a 3D culture model as well as in vivo. For instance, using their optimized mRNA:NP ratio and amount on tendon cells that are embedded in 3D constructs as mentioned in the Gehwolf et al. paper. Will the cells be able to take up the NPs as efficiently when under static tension and surrounded by collagen? Applying the current findings to more physiologically relevant models will provide important data. Other than these, there are a few minor comments listed below:
Specific Comments:
- There seems to be a word missing on page 2, second paragraph, line 3 “a biologically active molecule with a short half-life can be paired with an inactive (???), or”
- On page 3, section 2.2, line 1, please add abbreviated form of chemically modified mRNA in parentheses before using it further along in the text.
- On page 4, section 2.3.4, the cell isolation medium is different from the paper cited, (Gehwold et al). That study uses a more complex complete medium (see below) but does not contain Glutamax. In the interest of reproducibility, if using that exact method, please include the missing reagents. If only partially using that protocol, please say “adapted the protocol used by Gehwolf et al.”.
“The standard medium used is Minimum Essential Medium Eagle Alpha Modification (Sigma-Aldrich-Aldrich; #M4526) supplemented with 10% FBS, 100 units/ml penicillin, and 0.1 mg/ml streptomycin (Sigma-Aldrich-Aldrich, #P4333). The complete cell culture medium further contains 0.05 mM l-proline (1000× stock solution; Sigma-Aldrich-Aldrich, #P0380), 0.2 mM ascorbic acid (100× stock solution; Sigma-Aldrich-Aldrich, #A8960), and 10 μg/ml aprotinin (100× stock solution; Sigma-Aldrich, #A6106).” – from Gehwolf et al.
- In section 2.5, authors state “a solution of mRNA prepared in…” but previously use cmRNA for chemically modified RNA. If the solution contains cmRNA, kindly use the same abbreviation everywhere.
- It would be nice for the authors to address the immunogenicity of NPs and usage for rheumatoid arthritis treatments since it is one of the diseases mentioned early in the introduction.
Round 2
Reviewer 1 Report
Thanks for addressing my comments - there are still a few typos that need to be corrected.